# Consumer-based actions to reduce plastic pollution in rivers: A multi-criteria decision analysis approach

Luca Marazzi [1]*, Steven Loiselle[1,2], Lucy G. Anderson[3], Stephen Rocliffe[4], Debbie J. Winton[1]

**1** Department of Science Policy and Innovation, Earthwatch Institute (Europe), Oxford, United Kingdom, **2** Dipartimento di Biotecnologie Chimica e Farmacia, University of Siena, Siena, Italy, **3** Independent Research Consultant, Bath, United Kingdom, **4** College of Life and Environmental Sciences, University of Exeter, Penryn, United Kingdom

* lucamarazzi78@gmail.com

**Data Availability Statement:** Marazzi, L., S. Loiselle, L. Anderson, S. Roclifffe, and D. Winton. 2020. Data to support "Plastic Rivers project: Consumer-Based Actions to Reduce Plastic

## Abstract

The use and management of single use plastics is a major area of concern for the public, regulatory and business worlds. Focusing on the most commonly occurring consumer plastic items present in European freshwater environments, we identified and evaluated consumer-based actions with respect to their direct or indirect potential to reduce macroplastic pollution in freshwater environments. As the main end users of these items, concerned consumers are faced with a bewildering array of choices to reduce their plastics footprint, notably through recycling or using reusable items. Using a Multi-Criteria Decision Analysis approach, we explored the effectiveness of 27 plastic reduction actions with respect to their feasibility, economic impacts, environmental impacts, unintended social/environmental impacts, potential scale of change and evidence of impact. The top ranked consumer-based actions were identified as: using wooden or reusable cutlery; switching to reusable water bottles; using wooden or reusable stirrers; using plastic free cotton-buds; and using refill detergent/ shampoo bottles. We examined the feasibility of top-ranked actions using a SWOT analysis (Strengths, Weaknesses, Opportunities and Threats) to explore the complexities inherent in their implementation for consumers, businesses, and government to reduce the presence of plastic in the environment.

## Introduction

Managing plastic pollution has become a major international environmental priority [1] due to observed and estimated damage that plastics can cause to aquatic wildlife and ecosystems (both freshwater and marine) [2]. Between 4.8 and 12.7 million tonnes of plastic waste ends up in the ocean annually [3], the majority of which has land-based origins and is transported primarily by rivers [4, 5]. Billions of single-use plastic items are used annually in the UK alone (e.g. 14.5 plastic bottles [6]) and varying proportions of these are littered (e.g. 31.9% of cigarette butts are littered in the UK [7]). Littering can be defined as the intentional or

Pollution in Rivers: a Multi-Criteria Decision Analysis Approach" ver 1. Environmental Data Initiative. https://doi.org/10.6073/pasta/bed04a69d582bed901bc442078ce27c6 (Accessed 2020-07-13).

**Funding:** Lucy Anderson and Steve Rocliffe have been working on (the research underpinning, as per the respective roles indicated in our original submission) this manuscript as external consultants paid by Earthwatch Europe. This commercial relationship does not alter our adherence to PLOS ONE policies on sharing data and materials. Earthwatch Europe (in particular the co-authors Debbie Winton, Steven Loiselle and Luca Marazzi) played a role in the study design, data collection and analysis, decision to publish, and preparation of the manuscript. No external funding was received to conduct this study.

**Competing interests:** Lucy Anderson and Steve Rocliffe have been working on (the research underpinning, as per the respective roles indicated in our original submission) this manuscript as external consultants paid by Earthwatch Europe. This commercial relationship does not alter our adherence to PLOS ONE policies on sharing data and materials. Earthwatch Europe (in particular the co-authors Debbie Winton, Steven Loiselle and Luca Marazzi) played a role in the study design, data collection and analysis, decision to publish, and preparation of the manuscript. Steven Loiselle is a member of the editorial board of PLOS ONE; this did not have any bearing on our decision to submit to PLOS ONE and we are confident that the review process will not be affected either way by this.

unintentional discarding of any material into the environment and is a form of illegal behaviour that negatively affects society and environments from the local to the global level [8]. Although most marine plastics originate from land-based communities in river basins, plastic pollution in freshwater ecosystems is far less studied [9, 10]. Plastic pollution in rivers has been perceived by the public as posing higher immediate environmental risk than plastic pollution in open oceans, at least in inland communities [11]. Given that public and policy awareness is high, this is an opportune time to promote effective actions to reduce the flow of single-use plastic items entering the environment, by businesses, governments and the public. However, to guide people to tackle this problem, a complex and often conflicting array of recommendations is provided to the public, most of which are based on a limited understanding of the problem [12].

People's individual behaviours and lack of sufficient facilities (e.g. litter bins) are among the key factors causing the high volume of plastic litter in the environment. Negative individual behaviours, such as throwing cigarette butts in the street, are often widely adopted and form commonly accepted social practices (more informally, habits). Therefore, it is necessary for a large percentage of people to change behaviour to produce significant shifts in social practices, for example, a reduction or elimination of the widespread use of single-use water bottles. Groups of individuals, typically seen as consumers of market products (including plastic items), develop social practices and norms as determined by their individuality and competencies and by environmental, historic, cultural, social, and economic factors [13]. People who contribute the most to plastic pollution, for example by careless littering, may be less connected with both the issue itself (i.e. they ignore plastic pollution for lack of information or choice) and the location of accumulation (e.g. on roadsides, beaches). On the other hand, a growing number of people rate plastic pollution as a priority environmental concern [14, 15]. Therefore, increased awareness and sense of responsibility towards solving the plastic pollution problem should stimulate improved social practices. Over the past five years, plastics-related policy changes have increased at both the international (e.g. European Directive on single-use plastics [16]) and national level [17,18]. This heightened policy attention has been accompanied by an increase in business actions and environmental advocacy. Mixed coalitions of businesses and NGOs and other initiatives have been launched, for example the UK Plastics Pact [19], the Plastic Pollution Coalition [20], and the New Plastics Economy [21].

The recent increased awareness of plastic pollution and desire for action is inhibited by the inconsistent, if not conflicting, information provided to the public on how individuals can reduce this problem. Members of the public are faced with an array of opportunities to reduce and improve their plastic use and disposal practices; from brand selection to more environmentally friendly daily choices and actions. This contrasts with the need for consistent recommendations that are based on quantitative and reliable evidence to support their widespread adoption and activation. For example, the vast array of recycling symbols represents a persistent barrier to successful recycling [22]. Moreover, consumers may be unwilling to inform themselves on the relative environmental and economic benefits to individual actions. Sound, clear, and well-communicated information on, for example, the costs and benefits of alternative products is required and focused research to discern and implement the most effective plastic-reduction actions can help overcome present barriers [23].

Actions by consumers play an important role in promoting change and supporting effective environmental action [24]. However, there are limited studies examining which consumer actions are most impactful in reducing plastic pollution in aquatic ecosystems. Such baseline information is fundamental to favour effective changes in consumption-related decisions [24, 25]. Building on the findings of Winton et al. [10], which identified numerically dominant plastic items present in freshwater ecosystems in Europe, we evaluate consumer-based actions

to reduce or better manage single-use plastic items. Our focus on single-use plastics is due to their prevalence in freshwater ecosystems and to the fact that many single-use plastics are non-essential or unnecessary. These actions are prioritised using a multi-criteria decision analysis (MCDA). Multi criteria decision analysis is used widely in health, social and environmental sciences, for example to assess ecosystem services and environmental issues related to waste management, water, air, energy and natural resources [26]. In an MCDA application to plastic waste disposal [27], a panel of environmental scientists and environmental engineers assessed plastics waste disposal options (i.e. landfilling, recycling, incineration and pyrolysis) in relation to their environmental and health impacts, financial costs, practical and legal considerations. To our knowledge, this is the first study that uses MCDA to contextualise and prioritise actions related to plastic pollution. We combine MCDA with a SWOT (strengths, weaknesses, opportunities and threats) analysis to explore the merits and challenges linked to the implementation of specific priority actions. Our findings provide guidance for consumer action and insights for more effective government as well as corporate policy decisions to reduce plastic pollution, for example in terms of opportunity structures to encourage behavioural change.

## Methods

Our approach comprises a literature review to support the identification of relevant plastic reduction actions, a quantitative analysis and a qualitative evaluation of these actions (Fig 1).

### Data sources and literature review

To identify the best actions for individuals to reduce their plastic use and consequent pollution through littering or mismanaged plastic waste, we conducted a literature synthesis in October 2018. Relevant data from the published and grey literature were identified using a systematic search method to identify and evaluate interventions currently being recommended to consumers and businesses to minimise plastic waste. We identified 27 recommended actions to reduce plastic pollution by reviewing 187 papers, six books listed in Amazon's top 100

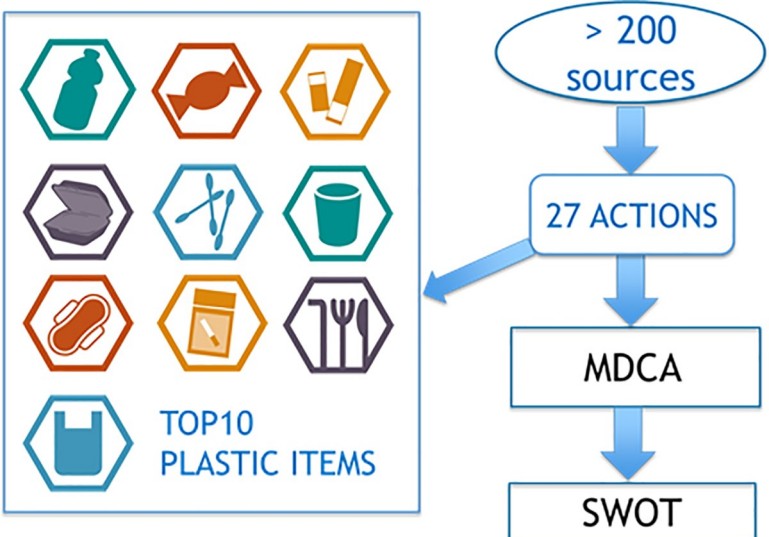

**Fig 1. Summary of our analysis approach: Multi-Criteria Decision Analysis (MCDA), Strengths Weaknesses Threats and Opportunities (SWOT).**

environmental non-fiction list, policy reports and NGO sources discussing plastic-reduction actions (see S1 Table in S1 File).

Business and consumer actions were identified and collated into a central database and selected based on their relevance to the top ten items identified by Winton et al. [10]. These top ten plastic items represented 42% of all litter identified in freshwater environments: five plastic items were food related (food wrappers, straws/stirrers/cutlery, bottles and lids, take-away containers, and cups), two were sanitary/cosmetic (cotton bud sticks and sanitary towels), and two were smoking related (cigarette butts and smoking-related packaging); the other item type was plastic bags.

## Multi-criteria decision analysis (MCDA)

We used MCDA to rank actions based on their positive and negative features. Multi-criteria decision analysis is an analytical method used to evaluate alternative decision options based on a set of common criteria [28, 29] and can be used to handle incomplete and uncertain information in a robust and flexible manner [30, 31]. We used MCDA to evaluate alternative consumer-based actions against practical, environmental, and economic aspects (these actions have management and policy implications relevant for various private and public stakeholders). The main steps of the process were: (1) problem formulation to identify ranking criteria to be evaluated; (2) assignment of weights to determine the overall relative importance of individual criteria; (3) review and scoring of each action against all criteria; (4) calculation of the total action score; (5) examination of the results and consistency checks [32]. Here we describe the different phases in detail.

(1) We identified a set of criteria against which to evaluate the actions to reduce the top ten plastic items identified in European freshwater environments [10]. These criteria focused on the feasibility, economic impacts, environmental impacts and unintended environmental consequences, potential scale of change, evidence of impact for each item with specific aspects taken from related studies.

(2) Twelve professionals working on plastics pollution in research, engagement or business and three authors (i.e. LM, SL, and DJW) assigned a weight (%) to each criterion based on a total weight of 100% per expert. The median weights were calculated for each criterion and the uncertainty around each median weight was determined using the interquartile range (Table 1). These twelve professionals (see acknowledgments) gave their consent to participate in the study in writing.

(3) We assigned scores, from 1 to 5, to each action based on available data related to their immediate and potential environmental impact, for example, the volume of different single-use items potentially reduced by that action (S2 Table in S1 File). Higher scores represent actions that have a higher positive impact (e.g. feasibility) to reduce plastic pollution or lower financial or environmental impacts (Table 1). Where no prior research was available, we assigned a score of '1' (very low).

(4–5) We multiplied each criterion's mean % weight by each criterion's 1–5 score and then summed all weighted scores to obtain a total score for each action (1).

$$Total\ action\ score = \sum_{i}^{10} score\ i * weight\ i \tag{1}$$

Where sufficient data were available, we also estimated the impact of each action in tonnes of plastic prevented from entering the environment annually, by multiplying the known annual number of items littered, or the annual number of items produced or disposed, by the average weight of individual plastic items, or of the plastic component of a multi-material item. Data on rates of littering were not available for bags, cups and straws and data on the weight of plastic within the item was not available for food wrappers. Therefore, we could not estimate the volume of plastic prevented from entering the environment for these items.

**Table 1. MCDA criteria with associated scoring system and expert-determined median weightings (i.e. relative percentage importance among criteria), with their relative spread (i.e. ± interquartile range).**

| Criterion | Weight (%) | Score (rank of 1 to 5) |
|---|---|---|
| **FEASIBILITY** | | |
| Likelihood of consumers performing the action *(how difficult it is for the consumer to make this change in their day to day life)* (LIK) | 16.0 ± 16.5 | 1 (very low—no systems in place) to 5 (very high—can add to existing recycling etc.) |
| Immediacy of the action (IMM) | 10.0 ± 5.8 | 1 (unavailable) to 5 (available immediately) |
| **ECONOMIC IMPACT** | | |
| Financial impact on consumer (CON) | 10.5 ± 8.5 | 1 (very high) - 5 (very low) |
| Financial impact on infrastructure or business (BUS) | 5.0 ± 6.9 | 1 (very high) - 5 (very low) |
| **ENVIRONMENTAL IMPACT** | | |
| Carbon impact for consumer (CAR1) *(e.g. from washing a reusable product at home)* | 5.0 ± 3.0 | 1 (very high) - 5 (very low) |
| Carbon impact of production (CAR2) *(e.g. from the product manufacturing process)* | 5.0 ± 4.7 | 1 (very high) - 5 (very low) |
| Water consumption impact (WAT) through product manufacturing and consumer use | 5.0 ± 6.3 | 1 (very high) - 5 (very low) |
| **OTHER ENVIRONMENTAL UNINTENDED CONSEQUENCES** | | |
| Other impacts specific to the action that need to be factored in *(e.g. ecological impact of harvesting an alternative material)* (UNI) | 10.0± 5.6 | 1 (very high) - 5 (very low) |
| **POTENTIAL SCALE OF CHANGE** | | |
| Reduction in the volume of plastic litter entering the environment (RED) | 10.0 ± 11.1 | 1 (very low) - 5 (very high) |
| **EVIDENCE OF IMPACT** | | |
| Examples showing potential effectiveness (e.g. pilot study/use in a particular country or region) (EGS) | 8.0 ± 5.0 | 1 (very low) - 5 (very high) with descriptive examples provided |

The sum of medians, differently from that of means, is not necessarily 100%.

Finally, we checked the total action score calculation and included a measure of uncertainty associated with the variability between different weights assigned to the ten criteria.

## SWOT analysis

To further evaluate each of the 27 plastic reduction actions identified, we used a qualitative SWOT (strengths, weaknesses, opportunities, and threats) analysis. This methodology is a strategic planning technique, originally employed by businesses to enhance their competitiveness [33], but more recently used to analyse environmental actions [34, 35]. We used SWOT analysis to appraise whether key aspects of each individual, business, or policy action may facilitate (strengths and opportunities) or hinder (weaknesses and threats) positive plastic pollution reduction. Whereas strengths and weaknesses are linked to current advantages and disadvantages of the various actions, opportunities and threats are linked to potential positive and negative aspects of their implementation. The analysis considered how each action influences both internal conditions of the system and the larger external context [36]; in our case, current market trends of commonly used plastic items and relevant legislation represent internal and external conditions, respectively. Strengths included, for example, the availability of alternative products on the market, or the presence of incentives for a consumer to adopt a more environmentally sustainable behaviour. Weaknesses included disadvantages for consumers, such as higher costs of alternative products, or inconveniences (e.g. carrying cups, boxes and cutlery for takeaway food and drinks).

## Results

Our findings comprise of quantitative results from the MCDA total action scores for the 27 plastic reduction actions (section 3.1) and SWOT qualitative results on strengths, weaknesses, opportunities and threats of the same actions (section 3.2).

## Quantitative ranking of plastic reduction actions: MCDA results

The weights given by the experts had a wide range (5–16%) and were not-normally distributed, in particular the weights of 'Likelihood of consumers performing the action (LIK)' and the estimated 'Reduction in the volume of plastic litter entering the environment' (Table 1). This wide range of weights reflects the diversity of experts consulted. The LIK criterion was given the highest weight (16%), the two financial impacts (CON and BUS) accounted for a total of 16%, as compared to a total of 25% for carbon (CAR1 and CAR2), water (WAT) and other environmental impacts (UNI). Cumulative weights of ~ 60% were given to socio-economic criteria (i.e. feasibility, economic impacts, scalability, and evidence of impacts) as compared to ~25% given to environmental criteria (i.e. carbon emissions, water use and other environmental unintended consequences).

A number of actions present feasibility, financial, and/or environmental issues. Consumers are less likely to carry reusable straws or seek products that are not yet widely available, such as drinks in recycled plastic bottles (LIK: scores 1–2). About two thirds of the actions were scored as immediately available, however various alternatives to plastic bottles are not (IMM: score 2; Table 2). Milk delivery, reusable glass coffee cup, and reusable straws had low total action scores because of their high financial costs for consumers and/or businesses (CON & BUS: scores 1–2; Table 2). Refilling detergent/ shampoo bottles, using solid soap, shampoo and conditioner bars (i.e. carbon emissions due to travelling to shops), using reusable nappies and reusable wet wipes (i.e. carbon emissions due to the washing required) had low total action scores because of their high carbon impacts on the consumer side (CAR1: scores 1–2; Table 2). The correct disposal of food wrappers, cigarette butts and smoking-related packaging had low total action scores, alongside the use of reusable bamboo coffee cup, menstrual cups and biodegradable wet wipes, and of reusable cotton tote bags because of these actions' high carbon impacts on the business side (linked to production process, disposal, and waste management) (CAR2: scores 1–2; Table 2). Water consumption impacts were deemed high (WAT: scores 1–2) for two thirds of the actions; while other unintended negative environmental consequences were high (UNI: scores 1–2) for the correct disposal of cigarette butts and smoking-related packaging and for the use of reusable and biodegradable wet wipes (e.g. the washing process consumes electricity and may release plastic microfibres).

We estimated the potential reduction of plastic litter (in tonnes/yr) of the 18 actions for which we could find sufficient relevant data. Our analysis shows that, if everyone in the UK took all these actions, a total of nearly 64,000 tonnes of plastic could be prevented from entering the freshwater environment annually (accounting only once for the impact of using various reusable coffee cups or alternative wet wipes, and other actions that can reduce plastic litter in non-additive ways) (Table 2). The actions with the highest plastic pollution reduction potential (tonnes/yr) were 'Use reusable nappies' (28,950 tonnes/yr), use a 'Reusable cotton tote bag' (9,000 tonnes/yr), 'Not flushing wet wipes' (3,400 tonnes/yr), and 'Reusable water bottle of any type' (6,741 tonnes/yr) (Table 2). The actions related to the ten dominant plastic items alone could reduce the release of plastic into the freshwater environment by ~ 25,000 tonnes annually (39% of our total estimate) (S2 Table).

In addition to the 27 actions discussed above, we identified 24 actions to which we could not assign scores due to a lack of data. Six of these additional plastic reduction actions (on food wrappers, cigarettes, cotton buds, and straws, stirrers and cutlery) are likely to be assessed in the near future (S3 Table in S1 File), while 18 actions are not because of a severe lack of data on, for example, their relative water and carbon impacts (S4 Table in S1 File). The total scores of the 27 actions ranged between 203 and 387 (Table 2). Six of the top ten most polluting consumer-related plastic items [10] were represented within the ten highest scoring actions.

**Table 2. Quantified effectiveness of consumer actions (total, weighted MCDA scores) to reduce plastic pollution related to the top ten most prevalent consumer macroplastic items in European freshwater environments [10]**

| Plastic Item | Actions | LIK | IMM | CON | BUS | CAR1 | CAR2 | WAT | UNI | RED | EGS | Total Score (weighted) | Overall rank of action | Potential impact (tonnes of plastic saved / year |
|---|---|---|---|---|---|---|---|---|---|---|---|---|---|---|
| Plastic bottles (including plastic lid or bottle top) | **Reusable water bottle of any type** | **5** | **5** | **5** | **4** | **3** | **4** | **5** | **5** | **5** | **5** | **386 ± 15** | **2** | **6,741** |
| | Milk delivery to home | 3 | 5 | 2 | 2 | 5 | 4 | 5 | 3 | 5 | 1 | 287 ± 60 | 15 | Unknown |
| | **Refill detergent/ shampoo bottles** | **3** | **2** | **5** | **3** | **2** | **5** | **5** | **5** | **5** | **1** | **333 ± 26** | **7** | **Unknown** |
| | Drinks in recycled plastic bottles | 1 | 2 | 4 | 3 | 4 | 5 | 5 | 5 | 5 | 2 | 294 ± 77 | 12 | Unknown |
| | Solid soap, shampoo and conditioner bars | 3 | 4 | 2 | 3 | 2 | 5 | 5 | 3 | 5 | 2 | 274 ± 70 | 18 | Unknown |
| | Drinks in a cardboard container | 3 | 2 | 4 | 3 | 5 | 5 | 1 | 4 | 5 | 2 | 270 ± 28 | 19 | Unknown |
| | Drinks in a glass container | 3 | 2 | 3 | 2 | 5 | 5 | 1 | 4 | 5 | 2 | 244 ± 46 | 22 | Unknown |
| Food wrappers | Correct disposal | 4 | 4 | 5 | 2 | 5 | 1 | 1 | 3 | 2 | 5 | 265 ± 34 | 21 | Unknown |
| Cigarette butts | Correct disposal | 3 | 4 | 5 | 2 | 5 | 1 | 1 | 2 | 2 | 2 | 231 ± 46 | 23 | 2,482[1] |
| **Plastic Item** | **Actions** | **LIK** | **IMM** | **CON** | **BUS** | **CAR1** | **CAR2** | **WAT** | **UNI** | **RED** | **EGS** | **Total Score (weighted)** | **Overall rank of action** | **Potential impact (tonnes of plastic saved / year** |
| Food takeaway containers | Reusable takeaway container | 3 | 5 | 4 | 3 | 4 | 4 | 1 | 3 | 5 | 3 | 304 ± 29 | 11 | 1,290[1] |
| Cotton bud sticks | **Substitute plastic sticks with paper in cotton buds** | **5** | **5** | **5** | **5** | **5** | **5** | **1** | **4** | **2** | **5** | **362 ± 49** | **3** | **61[1]** |
| Cups | **Reusable plastic coffee cup** | **4** | **5** | **4** | **2** | **3** | **4** | **4** | **5** | **2** | **5** | **331 ± 16** | **6** | **1,500[2]** |
| | Reusable glass coffee cup | 3 | 4 | 2 | 2 | 3 | 4 | 3 | 5 | 2 | 5 | 289 ± 27 | 13 | 1,500[2] |
| | Reusable bamboo coffee cup | 3 | 4 | 3 | 2 | 3 | 1 | 1 | 5 | 2 | 5 | 274 ± 16 | 20 | 1,500[2] |
| Sanitary items (nappies, sanitary towels, tampons and wet wipes) | **Not flushing wet wipes** | **3** | **5** | **5** | **5** | **5** | **5** | **5** | **3** | **1** | **2** | **306 ± 33** | **10** | **3,400[1]** |
| | Menstrual cups | 2 | 4 | 4 | 4 | 4 | 1 | 1 | 4 | 5 | 4 | 296 ± 45 | 14 | Unknown |
| | Disposable organic cotton sanitary towels | 3 | 4 | 5 | 4 | 4 | 2 | 1 | 3 | 5 | 3 | 293 ± 50 | 16 | 4,599[1] |
| | Reusable nappies | 3 | 4 | 4 | 5 | 2 | 4 | 2 | 3 | 5 | 2 | 291 ± 27 | 17 | 28,950[2] |
| | Reusable wet wipes | 3 | 4 | 3 | 5 | 2 | 3 | 2 | 2 | 4 | 2 | 219 ± 25 | 24 | 3,400[1] |
| | Biodegradable wet wipes | 4 | 2 | 4 | 2 | 5 | 1 | 1 | 2 | 4 | 1 | 207 ± 22 | 26 | 3,400[1] |
| **Plastic Item** | **Actions** | **LIK** | **IMM** | **CON** | **BUS** | **CAR1** | **CAR2** | **WAT** | **UNI** | **RED** | **EGS** | **Total Score (weighted)** | **Overall rank of action** | **Potential impact (tonnes of plastic saved / year** |
| Smoking related packaging | Correct disposal | 3 | 4 | 5 | 2 | 5 | 1 | 1 | 1 | 2 | 1 | 203 ± 45 | 27 | Unknown |
| Plastic straws, stirrers and cutlery | **Wooden cutlery** | **4** | **5** | **5** | **4** | **5** | **5** | **1** | **5** | **5** | **5** | **387 ± 24** | **1** | **222[1]** |
| | **Reusable cutlery** | **3** | **5** | **5** | **5** | **4** | **5** | **4** | **5** | **5** | **1** | **332 ± 41** | **8** | **222[1]** |
| | **Wooden stirrers** | **4** | **5** | **5** | **5** | **5** | **5** | **1** | **5** | **1** | **5** | **362 ± 53** | **4** | **0.2[1]** |
| | **Paper straws (recycle or compost after)** | **4** | **5** | **4** | **3** | **5** | **4** | **1** | **3** | **3** | **5** | **327 ± 43** | **5** | **2,533[2]** |
| | Reusable straws (bamboo/ steel/glass/ silicon) | 2 | 3 | 1 | 1 | 3 | 3 | 1 | 5 | 3 | 2 | 213 ± 56 | 25 | 2,533[2] |
| Plastic bags | **Reusable cotton tote bag** | **5** | **5** | **4** | **4** | **4** | **1** | **1** | **4** | **5** | **5** | **320 ± 35** | **9** | **9,000[2]** |

Acronyms for each category are listed in Table 1. Total median scores calculated using the criteria weights given by each of the 15 experts (± interquartile range). The top 10 actions are in bold. Potential impact calculated from the known percentage of the item that is littered ([1]) or the number of that item produced or disposed of per year ([2]).

However, the highest scoring actions of three of the top ten items (food wrappers, cigarette butts and cigarette packaging) were ranked in the bottom seven (Table 2) due to the limited availability of alternative actions, lack of data on carbon emission and water consumption impacts, and limited plastic pollution reduction potential (measured in tonnes/yr). Several actions had low scores because of the lack of available alternative products or options; in particular, reusable straws, recycled drink bottles (not yet widely available), and menstrual cups (LIK; Table 2). Twelve of the 27 actions relate to 'reuse' (44%), five to 'refuse' (19%), four to 'rot' (15%), three to 'recycle' (11%) and three to 'reduce' (11%) (S5 Table in S1 File).

## Qualitative evaluation of plastic reduction actions

The SWOT analysis indicated that, while most actions have at least four strengths, some actions have none; for example, the correct disposal of food wrappers and cigarette butts and smoking related packaging does not reduce the use of these items (S6 Table in S1 File, Fig 2).

Policy-related strengths included new EU legislation that will most likely reduce the use of plastic stirrers and cutlery and of plastic-based wet wipes, for example in parks near a river or on beaches (S6 Table in S1 File). Business-related strengths rely on the availability of alternative products or options while customer related strengths include, for example, the accessibility to products (e.g. wooden stirrers) or options (e.g. milk delivery) that reduce plastic use (S6 Table in S1 File).

| Strengths | Weaknesses |
|---|---|
| • Demand for more sustainable products is increasing<br>• Supply of reusable items, such as water bottles and coffee cups, is increasing<br>• New legislation in the EU and UK is likely to press businesses and public authorities to improve products and facilities<br>• Alternative items are functionally equivalent to ones made of plastics (e.g. wooden stirrers)<br>• Increased availability of facilities, such as refill stations and ballot bins, is likely to increase uptake of plastic reducing actions | • Weight, volume, and/or cost of alternative items make them inconvenient to purchase and then carry (e.g. reusable takeaway containers)<br>• People may forget reusable items and then purchase single-use items each time<br>• Lack of enforcement on anti-littering behaviours<br>• Lack of public funding for local councils to improve facilities (e.g. bins and refill stations)<br>• Environmental impact of alternative options may be higher (e.g. transporting glass bottles has higher carbon emissions)<br>• Alternative items might still be single-use and end up as litter |
| **Opportunities** | **Threats** |
| • New legislation and/or policy targets might help increase availability of reusable items (e.g. coffee cups) and/or specialized recycling schemes (e.g. for food wrappers or smoking packaging)<br>• Charities and other businesses could advertise or fundraise on reusable bottles, cups, or bags<br>• Availability and sustainability of biodegradable or reusable products could be increased by public and business support and policymaking and legislation<br>• Bans on single-use items could help spur the development of alternative products and/or behaviour changes (e.g. reduced use of non-essential items) | • People might become complacent about positive changes and ignore other changes they could make (i.e. moral licensing)<br>• People might discard paper/wood items in the environment as they perceive them as eco-friendly<br>• Greenhouse gas emissions from production and transport of some alternative materials (e.g. bamboo) may be higher than those of plastic items<br>• Demand for wood, paper, and cotton might unduly increase deforestation or environmental degradation |

**Fig 2. Synthesis of key findings from SWOT analysis (full results in S6 Table in S1 File).**

Policy-related weaknesses included a lack of facilities or incentives for people to switch to more sustainable actions. Business-related weaknesses, such as production cost or weight of items, point to a need for further innovation, and therefore link to related opportunities (S6 Table in S1 File). Consumer-related weaknesses include the weight and/or volume of alternative products to plastic items, such as metal water bottles or food containers (S6 Table in S1 File), which create inconveniences for consumers.

Opportunities on the policymaking side comprise the possibility for new legislation to reduce plastic content in cigarette filters, while opportunities for businesses and charities are related to the development and introduction of, and fundraising for, alternative products, such as reusable cups and bags (S6 Table in S1 File).

Threats were limited to unintended environmental impacts, for example increased land use for production of cotton for tote bags and moral licensing, whereby consumers may excuse themselves from additional, more onerous, actions (e.g. buy a reusable bottle) as they are already undertaking some simple actions to reduce their plastic use (e.g. refusing plastic straws and cutlery) (S6 Table in S1 File).

## Discussion

### Context and interpretation of our findings

The total action scores (Table 2) confirmed the value of a number of actions already underway in various countries (e.g. the UK ban on plastic straws, drinks stirrers and cotton buds [17]). Some of our top ten actions are among those already being adopted by consumers and businesses: replacing plastic cutlery and/or stirrers with wooden or reusable ones, using reusable water bottles, and using reusable water bottles (S2 Table in S1 File).

The 1–5 scores of actions against environmental criteria indicate that more data is needed on water consumption and carbon emission impacts of many actions (Table 1). Contrary to widespread beliefs, the production of many plastic products uses less water and generates lower carbon emissions than non-plastic alternatives [37–39]. Refusing or reducing single-use nonessential plastic items, such as plastic straws and heavily wrapped snacks, remains the simplest and highest-impact action that people can take to reduce their individual contribution of plastic pollution. Reducing the consumption of snacks, bottled fizzy drinks and cigarettes also have multiple health-related co-benefits [40]. Therefore, future studies and additional data will help reduce the uncertainty connected with the lack of information on, for example water and carbon impacts of different plastic reduction actions.

Consumer-based actions are limited for some of the key plastic items; for example, most consumers can only dispose correctly of food wrappers, cigarette butts and cigarette packaging to reduce plastic pollution. However, recycling schemes are emerging that can recycle food wrappers and cigarette waste (e.g. Terracycle®), although their economic feasibility and sustainability have not been well documented. While most actions are complementary to one another and they add up to reducing plastic waste and thus pollution, a few actions are alternative to one another because they deal with using a reusable product made from different materials (e.g. for straws and toothbrushes).

Working within these limited options, there is a clear need for (i) a general reduction of consumers' reliance on single-use plastic items; (ii) an increase in the availability of alternative reusable items to replace single-use plastic items at an affordable cost and with lower documented environmental impact. Given people's increased awareness of plastic pollution and legislation changes in various countries, stakeholders (e.g. policymakers and businesses) can take advantage of identified strengths to overcome current weaknesses, limit threats, and/or expand opportunities. As many of these weaknesses are cost-related (e.g. availability of in-

store reusable cups and takeaway containers), these consumer actions could be supported by related business actions. Clear incentives towards behaviour change are needed to support end-of-pipeline actions, such as proper disposal and management of plastic waste (e.g. wet wipes, smoking related litter). Ultimately, innovative approaches that attempt to improve personal appreciation for the environment can help increase the uptake of pro-environmental behaviours and may thus have more permanent effects than policies [41]. However, a central threat across all actions is the possibility that people may take a moral license to increase (or at the least, not reduce) plastic use in other areas of their lives. Such moral licensing could worsen overall plastic consumption, as perceived benefits from one action are offset by the elimination of different, perhaps more impactful, actions [42].

Creative initiatives have been shown to limit plastic littering, for example ballot bins whereby people are encouraged to vote by throwing their cigarette butt in the section of the bin with their preferred answer to a set question (e.g. about their sport teams, food or other preferences) [43]. However, such initiatives could end up incentivising the generation of more waste by creating the illusion that the plastic waste from cigarette butts is currently managed properly/recycled. Ballot bins cannot realistically be placed everywhere, but people may get complacent and relax their plastics-related standards in other parts of their daily routines.

Although MCDA is a commonly used method in various disciplines, it presents some limitations linked to, for example, the meaning of the weights and scores of the individual criteria, the validity of the aggregate multi-attribute value function, and the use of a single score to adequately characterize a complex situation [44]. Therefore, more work is needed to expand and strengthen the evidence base on the effectiveness, practicality, and environmental impacts of different plastic pollution reduction actions. This can be done by using alternative and/or complementary methodologies to MCDA (e.g. Life Cycle Assessment) to obtain additional environmental impact information [45], to assess unassessed actions (see S4 Table in S1 File), and to generate a broader picture of what people can do to reduce plastic pollution. Here, we excluded bioplastics (i.e. plastic materials produced from renewable biomass sources, such as vegetable fats and oils, corn starch, and recycled food waste) from our analysis of possible actions, as benefits and environmental costs of these materials have yet to be fully determined [46]. It is therefore a priority to generate new insights into viability and sustainability of alternative products, materials, and options so that researchers, policymakers, and citizens can all do their part in tackling the growing environmental and societal challenge represented by plastic pollution.

## Future research directions and policy implications

Our results indicate several "blockers" that consumers encounter in attempting to reduce their plastic pollution through their choice to purchase, use, and manage plastic (e.g. costs, inconvenience, time constraints, lack of available facilities or customer support). One key obstacle is the lack of clarity on: (i) proper recycling practices for the wide variety of available plastic products; (ii) the best functionally equivalent alternatives to plastic items based on comparative carbon and water impact estimates; (iii) which items are truly compostable or biodegradable. In Europe, there are too many different recycling arrow-like symbols on plastic packaging (e.g. "widely recycled" or "check with your local council"), which end up confusing people [22]. Simpler and clearer recycling labelling systems could create confidence and participation among consumers so they can recycle more plastics and contribute to addressing this challenge.

Governments are already acting to reduce plastic pollution. For example, more than 60 countries have introduced bans and levies to curb single-use plastic waste [47]. Various

supermarkets and other businesses have reduced the number of plastic bags and packaging they provide their customers with [48]. To implement the recommended actions (Table 2), financial incentives could be given to improve consumer uptake and help businesses enable consumer action (e.g. provide takeaway reusable containers in restaurants) (see S6 Table in S1 File). Such incentives could come from local, regional, or national government or business associations to support individual and collective action. An increase in related infrastructure, such as more and/or more visible water refill stations in crowded places, such as railway stations and shopping malls, would also help reduce plastic pollution by encouraging people switch from single-use to reusable water bottles.

Excessive reliance on single-use plastics can be reduced by looking at individual behaviours and how these spread between people within and across groups and society. Routinized activities that people perform in their everyday lives constitute social practices that stem from widespread individual behaviours determined by cultural, material and competence factors [13]. However, social practices are not immutable. For example, a preferred commuting mode is both a recognizable pattern and a performance that can be changed, albeit with some effort required to explore alternative transportation modes, such as cycling [13]. People may experiment plastic-clever actions that may be seen as odd by their peers because they do not conform to common and accepted behaviours (e.g. holding a balloon-free party for children [49]). After initial attempts, the motivation of such pioneers to reduce their environmental impact may decline, especially in the absence of positive pressure from peers or friends and family. Policies to prevent littering and incorrect disposal, reduce plastic use, and improve plastic waste management, need to encourage and sustain individual actions and help these become mainstream (i.e. the new norm).

To enact systemic change in the way consumer plastics are used and disposed of, proactive informed consumer action will play an important role. Having now identified what people can do to reduce plastic pollution, we need to continue to ask: (1) What are the most popular and feasible actions that people already take in the home and when they are on-the-go (e.g. commuting or travelling) to reduce their plastic footprint?; (2) What are the challenges or barriers to reducing plastic use, especially single-use items?; (3) Which actions taken at individual and household levels are having the biggest impact in reducing plastic use, especially of single-use items? To address questions (1) and (2), Earthwatch Europe has developed an online plastic footprint calculator to identify barriers to reducing plastic use in people's activities on-the-go, i.e. while they commute, run their daily errands, and live their social life out of their home [50]. Preliminary findings from about 1,200 respondents (the great majority living in the UK) suggest that people are most concerned and active in trying to reduce their use of food wrappers, sanitary items, and single-use takeaway containers.

Studies on drivers, sources and pathways of plastic pollution need to be integrated with studies on people's behaviours and social practices. Findings from such integrated work can help reduce the use and improve the disposal of various plastic items used in the household and on-the-go. These improvements are particularly urgent in areas that may have already been identified as plastic litter hotspots, for example by local community groups or environmental NGOs. To tackle knowledge gaps on patterns and drivers of people's behaviours, social scientists can help better understand key factors that affect behaviour change and evolution of social practices. These should include studies on social dynamics, moral norms, identity, awareness of consequences, ascription of responsibility, perceived and actual behavioural control, knowledge and specific attitudes [10]. Baseline information on consumer behaviour around purchasing and use of alternative products is needed, upon which strategies for change can be built. Consumers need evidence-based advice that is simply presented and can be easily put into practice in their everyday lives. Standardised comparisons of environmental and

socio-economic benefits and disadvantages of different and potentially complementary actions (e.g. reduce plastic water bottles and buy a reusable bottle) are urgently needed. The availability of well-researched, clearer and simplified information would focus the general public's attention on local pollution issues and support legislation, novel recycling technologies and product innovation. Increased public awareness based on sound scientific evidence can boost individual and collective action to reduce plastic pollution [51]. For example, education programs have shown to successfully reduce waste in Europe [52], Chile [53] and Australia [8] by creating a sense of environmental responsibility in participants [53].

The positive outcomes of the numerous social enterprises and campaign groups tackling plastic pollution globally [8] are already being multiplied through communities and social media. Influencers and neighbourhood incubators can help people and groups spread the message and imitate one another to achieve positive change on a larger scale [54].

## Conclusions

The general recommendations for actions to reduce plastic pollution that emerged from the present study were: (1) refuse non-necessary plastic items, such as straws; (2) reduce dependence on traditionally single-use plastic items (e.g. shampoo bottles), for example by refilling or buying larger bottles; (3) replace plastic items with reusable and/or alternative products with a lower environmental impact; (4) correctly dispose of items, such as wet wipes, that may be essential and thus impossible to refuse or reuse. Refusing problematic plastic items remains the most direct action to lower plastic pollution levels without negative consequences. Some of our recommendations are not universally applicable as, for example, plastic straws and other products may be needed for medical reasons.

The path from awareness raising to action can be tortuous, with psychological and practical barriers that often block widespread behaviour change in our societies [55]. People often deny the damage that buying and shortly after throwing away large numbers of single-use plastic bottles or bags causes to the environment and/or they deny their share of responsibility [10]. However, people can change their behaviour to be more responsible towards the environment by acting in a self-determined way, for example by estimating costs and identifying co-benefits of reducing plastic use. Younger people who are still in the process of habit formation, can exert a particularly important positive impact on their peers and, with time, on the next generations. More and/or better information on plastic use and disposal is likely to help, but may not be sufficient to motivate people to do their part to reduce plastic pollution in rivers. Users of plastic items belong to different social groups and face different situations in their everyday lives, depending on their socio-demographic and socio-cultural situation. Some may need to absorb and internalize the information received or consult peers or leaders they trust, such as friends, colleagues, family members, or experts. Various competences and capacities (e.g. financial) and material structures are needed to support the use of reusable products, or the uptake of alternative materials to plastics [13].

Behaviours such as buying a reusable bottle need to be made easy and practical for people of all ages and backgrounds so that they can become the most logical and convenient options. Credible alternative actions and products offered by businesses or through legislation at competitive costs can enable positive behaviour changes that ultimately reduce plastic pollution (e.g. reusable products). Every person, business, authority and organisation have a role to play to reduce plastic use, incorrect disposal and thus pollution. More environmentally sustainable choices and actions can become new norms and widely accepted social practices, if robustly supported by private and public sector initiatives, well-enforced policies, and evidence-based media reporting. Our study demonstrates that consumers-citizens can greatly contribute to

solving the plastic pollution problem and can be used as a stepping stone for further interdisciplinary research.

## Supporting information

**S1 File.**
(DOCX)

## Acknowledgments

We gratefully acknowledge the input from current and former Earthwatch Europe colleagues (Jacqualyn Eales, Josephine Head, Gitte Kragh, Stephen Parkinson, Justin Robinson, Paul Scott, Kesella Scott Somme) and external experts (Lesley Henderson, Sherri Mason, Solaja Oludele, Teresa Perez, Rosie Tudor) to assign percentage weights to the criteria used to rank plastic reduction actions. We also thank Johanna Ritter (Institut für Sozialinnovation, Berlin) and Dilyana Mihaylova (Plastic Oceans UK) for providing insightful comments on drafts of the manuscript.

## Author Contributions

**Conceptualization:** Steven Loiselle, Lucy G. Anderson, Stephen Rocliffe, Debbie J. Winton.

**Data curation:** Luca Marazzi, Lucy G. Anderson, Stephen Rocliffe, Debbie J. Winton.

**Formal analysis:** Luca Marazzi, Lucy G. Anderson, Stephen Rocliffe, Debbie J. Winton.

**Investigation:** Luca Marazzi, Steven Loiselle, Lucy G. Anderson, Stephen Rocliffe, Debbie J. Winton.

**Methodology:** Lucy G. Anderson, Stephen Rocliffe, Debbie J. Winton.

**Supervision:** Steven Loiselle.

**Validation:** Steven Loiselle.

**Writing – original draft:** Luca Marazzi, Steven Loiselle, Debbie J. Winton.

**Writing – review & editing:** Luca Marazzi, Steven Loiselle, Lucy G. Anderson.

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
