## [Decision Letter · Decision Letter 0]

9 Mar 2020

PONE-D-20-03363

Consumer-Based Actions to Reduce Plastic Pollution in Rivers: a Multi-Criteria Decision Analysis Approach

PLOS ONE

Dear Dr. Marazzi,

Thank you for submitting your manuscript to PLOS ONE. After careful consideration, we feel that it has merit but does not fully meet PLOS ONE’s publication criteria as it currently stands. Therefore, we invite you to submit a revised version of the manuscript that addresses the points raised during the review process.

We would appreciate receiving your revised manuscript by Apr 20 2020 11:59PM. To enhance the reproducibility of your results, we recommend that if applicable you deposit your laboratory protocols in protocols.io, where a protocol can be assigned its own identifier (DOI) such that it can be cited independently in the future. For instructions see: http://journals.plos.org/plosone/s/submission-guidelines#loc-laboratory-protocols

We look forward to receiving your revised manuscript.

Kind regards,

Vassilis G. Aschonitis

Academic Editor

PLOS ONE

Journal Requirements:

2. Please provide additional information on participant consent to take part in the study, and whether consent was written or verbal. We also note that some of the participants appear to be named in the Acknowledgements - please clarify whether these individuals explicitly consented to be named in the Acknowledgements.

4.  Thank you for stating the following in the Financial Disclosure section:

"Lucy Anderson and Stephen Rocliffe were funded for this work by Earthwatch as external collaborators. ".

We note that one or more of the authors are employed by a commercial company: 'Earthwatch Institute and Independent research consultant, Bath'.

Reviewers' comments:

Reviewer's Responses to Questions

**Comments to the Author**

1. Is the manuscript technically sound, and do the data support the conclusions?

Reviewer #1: Yes

2. Has the statistical analysis been performed appropriately and rigorously? 

Reviewer #1: Yes

3. Have the authors made all data underlying the findings in their manuscript fully available?

Reviewer #1: Yes

4. Is the manuscript presented in an intelligible fashion and written in standard English?

Reviewer #1: Yes

5. Review Comments to the Author

Reviewer #1: The manuscript uses a multi-criteria decision analysis (MCDA) to contextualise and prioritise actions related to plastic pollution. In addition, plastic reduction consumer-based actions using SWOT (strengths, weaknesses, opportunities and threats) analysis were explored. The article is well organized and written and gives adequate references to related work. Figures and tables are helpful to resume the text. The goals of the study are clearly formulated, the technical procedures (MCDA analysis etc) are well described. The results are adequate and presented in a clear manner. Overall, the paper is of general interest. However, some points should be better explained.

Specific comments

• Did the authors consider any barriers or constrains of the for the MCDA? Please comment.

• Did the authors notice any conflicts in customer actions in the MCDA?

• Some data for uncertainty of the input data to the MCDA method selection rules should be added in the manuscript.

• The findings are not compared with similar studies in the open literature. Please comment.

• Page, lines 212-235: this paragraph should be re-written by describing the scores with a more effective way.

• I would suggest the authors to include a figure or diagram describing the main steps of the methodology used.

6. PLOS authors have the option to publish the peer review history of their article (what does this mean?). If published, this will include your full peer review and any attached files.

Reviewer #1: No

---

## [Author Response · Author response to Decision Letter 0]

14 May 2020

Dear Dr. Aschonitis

Thank you again for the constructive review and the opportunity to submit our revised manuscript. In the following paragraphs, we describe the revisions made to our manuscript, PONE-D-20-03363, with respect to the comments and suggestions provided by the reviewer. We have followed all the suggestions and we thank the reviewer for their constructive comments and ideas to further improve our manuscript. We have also made some other minor changes to enhance the readability of the text; all the changes made are marked in the version with tracked changes.

Reviewer #1: The manuscript uses a multi-criteria decision analysis (MCDA) to contextualise and prioritise actions related to plastic pollution. In addition, plastic reduction consumer-based actions using SWOT (strengths, weaknesses, opportunities and threats) analysis were explored. The article is well organized and written and gives adequate references to related work. Figures and tables are helpful to resume the text. The goals of the study are clearly formulated, the technical procedures (MCDA analysis etc) are well described. The results are adequate and presented in a clear manner. Overall, the paper is of general interest. However, some points should be better explained.

Specific comments

1) Did the authors consider any barriers or constraints of the … for the MCDA? Please comment.

Authors’ response: Thank you for this suggestion. In the revised manuscript, we added a discussion point on how MCDA has limitations and other techniques could be used to integrate additional information in future studies.

“Although MCDA is a commonly used method in various disciplines, it presents some limitations linked to, for example, the meaning of the weights and scores of the individual criteria, the validity of the aggregate multi-attribute value function, and the use of a single score to adequately characterize a complex situation44. Therefore, more work is needed to expand and strengthen the evidence base on the effectiveness, practicality, and environmental impacts of different plastic pollution reduction actions. This can be done by using alternative and/or complementary methodologies to MCDA (e.g. Life Cycle Assessment) to obtain additional environmental impact information45, to assess unassessed actions (see Table S4), and to generate a broader picture of what people can do to reduce plastic pollution. Here, we excluded bioplastics (i.e. plastic materials produced from renewable biomass sources, such as vegetable fats and oils, corn starch, and recycled food waste) from our analysis of possible actions, as benefits and environmental costs of these materials have yet to be fully determined46. It is therefore a priority to generate new insights into viability and sustainability of alternative products, materials, and options so that researchers, policymakers, and citizens can all do their part in tackling the growing environmental and societal challenge represented by plastic pollution.”

2) Did the authors notice any conflicts in customer actions in the MCDA?

Authors’ response: We included a short discussion on how most actions are complementary to one another and they add up to reducing plastic waste. We note that some actions are alternative to one another because they relate to choosing between different types of non-plastic toothbrush or straw.

“Consumer-based actions are limited for some of the key plastic items; for example, most consumers can only dispose correctly of food wrappers, cigarette butts and cigarette packaging to reduce plastic pollution. However, recycling schemes are emerging that can recycle food wrappers and cigarette waste (e.g. Terracycle®), although their economic feasibility and sustainability have not been well documented. While most actions are complementary to one another and they add up to reducing plastic waste and thus pollution, a few actions are alternative to one another because they deal with using a reusable product made from different materials (e.g. for straws and toothbrushes).”

3) Some data for uncertainty of the input data to the MCDA method selection rules should be added in the manuscript.

Authors’ response: We included an improved explanation in the Methods and Discussion on how ranking scores 1-5 were determined (literature review), their uncertainty and the need for further study. 

[Methods]

(2) Twelve professionals working on plastics pollution in research, engagement or business and three authors (i.e. LM, SL, and DJW) assigned a weight (%) to each criterion based on a total weight of 100% per expert. The median weights were calculated for each criterion and the uncertainty around each median weight was determined using the interquartile range (Table 1). 

(3) We assigned scores, from 1 to 5, to each action based on available data related to their immediate and potential environmental impact, for example, the volume of different single-use items potentially reduced by that action (Table S2). Higher scores represent actions that have a higher positive impact (e.g. feasibility) to reduce plastic pollution or lower financial or environmental impacts (Table 1). Where no prior research was available, we assigned a score of ‘1’ (very low).

[Discussion]

“Although MCDA is a commonly used method in various disciplines, it presents some limitations linked to, for example, the meaning of the weights and scores of the individual criteria, the validity of the aggregate multi-attribute value function, and the use of a single score to adequately characterize a complex situation44. Therefore, more work is needed to expand and strengthen the evidence base on the effectiveness, practicality, and environmental impacts of different plastic pollution reduction actions. This can be done by using alternative and/or complementary methodologies to MCDA (e.g. Life Cycle Assessment) to obtain additional environmental impact information45, to assess unassessed actions (see Table S4), and to generate a broader picture of what people can do to reduce plastic pollution.”

4) The findings are not compared with similar studies in the open literature. Please comment.

Authors’ response: Our MCDA analysis was built upon the latest quantitative studies around individual consumer plastic waste items. We are not aware of a similar study comparing the relative opportunities and challenges around different consumer, business and policy actions to address consumer plastic waste, hence not referencing similar analyses in the discussion. We are aware of a recent MCDA study to evaluate end-of-life management options for plastic fishing gear (e.g. see Deshpande et al. 2020) and an opinion piece arguing the need for consumer, business and governance actions in a study of plastic waste in marine and coastal environments (see Vince & Hardesty, 2016). However, these studies do not quantify actions to reduce consumer plastic waste so it is not possible to make direct comparisons.

5) Page, lines 212-235: this paragraph should be re-written by describing the scores with a more effective way.

Authors’ response: We revised the paragraph and re-ordered Table 1 columns for better clarity.

“(3) We assigned scores, from 1 to 5, to each action based on available data related to their immediate and potential environmental impact, for example, the volume of different single-use items potentially reduced by that action (Table S2). Higher scores represent actions that have a higher positive impact (e.g. feasibility) to reduce plastic pollution or lower financial or environmental impacts (Table 1). Where no prior research was available, we assigned a score of ‘1’ (very low).”

6) I would suggest the authors to include a figure or diagram describing the main steps of the methodology used.

Authors’ response: Thank you for the suggestion. In the revised manuscript, we added a diagram of the main steps to the approach (Fig 1).

Figure 1. Summary of the different steps of our analyses: Multi-Criteria Decision Analysis (MCDA), Strengths Weaknesses Threats and Opportunities (SWOT)

---

## [Decision Letter · Decision Letter 1]

8 Jul 2020

Consumer-Based Actions to Reduce Plastic Pollution in Rivers: a Multi-Criteria Decision Analysis Approach

PONE-D-20-03363R1

Dear Dr. Marazzi,

We’re pleased to inform you that your manuscript has been judged scientifically suitable for publication and will be formally accepted for publication once it meets all outstanding technical requirements.

Kind regards,

Vassilis G. Aschonitis

Academic Editor

PLOS ONE

Additional Editor Comments (optional):

Reviewers' comments:

Reviewer's Responses to Questions

**Comments to the Author**

1. If the authors have adequately addressed your comments raised in a previous round of review and you feel that this manuscript is now acceptable for publication, you may indicate that here to bypass the “Comments to the Author” section, enter your conflict of interest statement in the “Confidential to Editor” section, and submit your "Accept" recommendation.

Reviewer #1: All comments have been addressed

2. Is the manuscript technically sound, and do the data support the conclusions?

Reviewer #1: Yes

3. Has the statistical analysis been performed appropriately and rigorously? 

Reviewer #1: Yes

4. Have the authors made all data underlying the findings in their manuscript fully available?

Reviewer #1: Yes

5. Is the manuscript presented in an intelligible fashion and written in standard English?

Reviewer #1: Yes

6. Review Comments to the Author

Reviewer #1: The manuscript has been improved taking into consideration the reviewers' suggestions. All comments have been addressed.

7. PLOS authors have the option to publish the peer review history of their article (what does this mean?). If published, this will include your full peer review and any attached files.

Reviewer #1: No

---

## [Editor Report · Acceptance letter]

6 Aug 2020

PONE-D-20-03363R1 

Consumer-Based Actions to Reduce Plastic Pollution in Rivers: a Multi-Criteria Decision Analysis Approach 

Dear Dr. Marazzi:

I'm pleased to inform you that your manuscript has been deemed suitable for publication in PLOS ONE. Congratulations! Your manuscript is now with our production department. 

Kind regards, 

on behalf of

Dr. Vassilis G. Aschonitis 

Academic Editor

PLOS ONE